# *T. gondii* Infection in Urban and Rural Areas in the Amazon: Where Is the Risk for Toxoplasmosis?

**DOI:** 10.3390/ijerph18168664

**Published:** 2021-08-17

**Authors:** Rafaela dos Anjos Pinheiro Bogoevich Morais, Ediclei Lima do Carmo, Wanda Silva Costa, Rodrigo Rodrigues Marinho, Marinete Marins Póvoa

**Affiliations:** 1Curso de Pós-graduação em Biologia de Agentes Infecciosos e Parasitários, Instituto de Ciências Biológicas da Universidade Federal do Pará, Belém 66075-110, Brazil; povoamm@gmail.com; 2Seção de Parasitologia, Instituto Evandro Chagas/SVS/MS, Ananindeua 67030-000, Brazil; edicleicarmo@iec.gov.br (E.L.d.C.); wandacosta@iec.gov.br (W.S.C.); rodrigomarinho@iec.gov.br (R.R.M.)

**Keywords:** *Toxoplasma*, prevalence, Amazon

## Abstract

Toxoplasmosis, an infection caused by *Toxoplasma gondii*, which is found worldwide, can affect human and animal health in different ways. This study aimed to estimate the infection prevalence in humans and to determine risk factors related to urban and rural areas in a municipality in the Brazilian Amazon where an outbreak had been registered. Blood samples for serological analysis were obtained, and interviews were performed to fill out an epidemiological questionnaire. A total of 1140 individuals were included, of which 70.6% (804/1140; 95% CI: 67.9–73.2%) were positive for IgG anti-*T. gondii* antibodies. In rural areas, the prevalence was 62.6% (95% CI: 58.9–66.3%), while in urban areas, it was 81.9% (95% CI: 78.4–85.4%). The risk of becoming infected in urban areas was 2.7 times higher (95% CI = 2.0–3.6%) than that in rural areas. When comparing the prevalence in the age group from 1 to 10 years in both areas, the rate was 28.6% (42/147; 95% CI: 21.3–35.9%) for rural areas and 69.4% (61/88; CI 95%: 59.7–79.0%) for urban areas. Therefore, it is concluded that parasite exposure starts in the first years of life in urban areas and that disordered urban area expansion may cause an increase in exposure to the different strains of *T. gondii* present in the Amazon.

## 1. Introduction

Toxoplasmosis, an infection caused by the protozoan *Toxoplasma gondii*, affects humans and other homeothermic animals. Transmission usually occurs by consumption of undercooked meat from infected animals or food/water contaminated by oocysts. On the other hand, vertical transmission can occur when a woman acquires the infection during pregnancy, causing congenital toxoplasmosis. Symptoms are normally characterized by fever, myalgia and adenomegaly, although some individuals, especially immunocompromised individuals, develop ocular involvement and splenomegaly. In congenital toxoplasmosis, central nervous system involvement can occur, resulting in ocular lesions and microcephaly [1]. In these cases, treatment is crucial, even in subclinical infections, for reducing the severity of the disease.

Infection affects one-third of the world’s human population, but the prevalence varies from region to region, depending on the climate and population habits. Prevalence rates of 3.4% to 17.5% have been reported in different regions of China [2], and rates of 11.4% have been reported in the United States [3]. In Brazil, it is estimated that 60% of the population is infected [4], and in the Amazon, 66.8% seroprevalence has been observed in indigenous people [5] and 78% in the city of Belém, State of Pará [6].

In the Amazon, the *T. gondii* cycle is very complex, as conditions such as temperature and humidity, diversity of animal species, water influence and interdependence between rural/wild and urban areas favor the dispersion of oocysts and the circulation of atypical strains; this implies a risk of developing so-called “Amazonian toxoplasmosis”, which causes serious manifestations and deaths in immunocompetent patients, as reported in Suriname and French Guiana [7].

In Brazil, outbreaks associated with the consumption of water and food contaminated with oocysts have been registered [8]. In addition, in the Brazilian Amazon, outbreaks of toxoplasmosis associated with açaí juice (fruit extracted from the forest and highly consumed in the region) intake have been recorded [9,10]. In 2013, an outbreak of toxoplasmosis was reported in Ponta de Pedras municipality, Pará State, Brazil, which included 90 individuals aged from 11 months to 64 years old. Infection of pregnant individuals and ocular and multivisceral toxoplasmosis were not detected. Epidemiological and spacial analysis suggested that the source of infection was açaí juice.

Depending on their type of exposure (animal contact, population density, types of food consumed, wild area contact, sanitation conditions), Amazon residents from urban and rural areas, including those from riverside communities (who live along rivers and live on fisheries and products extracted from the forest), are at risk of acquiring infection with different strains of *T. gondii* and developing different manifestations of toxoplasmosis. These important factors in addition to the distance to the reference health services can lead to a greater delay in diagnosis and timely treatment. Despite this, few studies have been carried out in these communities, resulting in a lack of information, which hampers the implementation of toxoplasmosis control strategies aiming at different landscapes and populations. Therefore, this study aimed to determine the prevalence and identify the risk factors for infection by *T. gondii* in rural and urban communities in a municipality in the Brazilian Amazon after an outbreak occurred.

## 2. Materials and Methods

### 2.1. Ethical Aspects

This study was approved by the Ethics Committee of Instituto Evandro Chagas on 26 August 2014 (CAAE 31630514.3.0000.0019/n° 764.785). The sample collection and technical procedures were only initiated after adult participants provided written informed consent and parents or guardians provided written informed consent for the participating children, in accordance with Brazilian regulations.

### 2.2. Study Area

The study was carried out in the municipality of Ponta de Pedras (01°42’ 3” S and 48°18’52” W), located in the Marajó Archipelago, State of Pará, Brazilian Amazon (Figure 1), with an area of 3,365,148 km^2^ and a territory divided into floodplain and upland areas. The estimated resident population is 25,999 inhabitants, 49% of whom live in the urban area, which is surrounded by rivers and forests, and the rest are distributed in rural areas (which include small villages on the mainland and riverside communities with residences distant from each other). Only 20% of the population has sewage and adequate sanitary treatment [11,12].

### 2.3. Study Design and Samples Collection

Considering a population of 25,999 inhabitants, with a regional seroprevalence of 65% [13] and adopting a 95% confidence level, 5% margin of error and 10% nonadherence to the study, a minimum sample of 364 individuals was calculated by the program Epi Info version 7.2.2.1 (Centers for Disease Control and Prevention (CDC Atlanta)).

From August 2015 to April 2018, a cross-sectional study was conducted with interviews and blood collection from 1140 individuals in health units and by active searches in residences in urban and rural areas. All participants were aged over one year and were residents of urban area neighborhoods or rural communities. Pregnant women were excluded from the study.

Approximately 5 mL of venous blood was collected from each individual, and serum aliquots were storage at −20 °C until serology testing.

Sociodemographic data and epidemiological information were obtained by interview, and information collected included place of residence, age, sex, type of water consumed, consumption of raw/undercooked meat and soil and animal contact. Data were entered in a database with Epi Info program version 7.2.2.1.

### 2.4. Serological Evaluation

Serum samples were tested for detection of *T. gondii* specific IgG by immunocapture immunoenzymatic assay (ELISA) (Symbiosis Diagnóstica Ltda., Leme, Brazil, which has a sensitivity and specificity of 99.46% and 99.16%, respectively), following the procedures and cutoff calculation of the manufacturer. In addition to the positive and negative controls available in the kit, laboratory reference controls were included.

### 2.5. Statistical Analysis

Descriptive and inferential statistical methods were used. Bivariate analysis (chi-square test) was used to identify the risk factors, and variables that had a *p*-value < 0.200 were performed by the stepwise forward method (multiple logistic regression). The factors that showed an association with the dependent variable (positive for anti-*T. gondii* IgG antibodies) were identified during the succession of steps. The data were analyzed with statistical programs Epi Info version 7.2.2.1.and BioEstat program version 5.3.

## 3. Results

The study included 469 (41.1%) individuals from the urban area and 671 (58.8%) from the rural area. The age of the population studied ranged from 1 to 89 years, with a mean age of 28.9 ± 6.0 (95% CI: 26.9–30.6). Most individuals were aged 11 to 20 years (21.5%, 245/1140), were female (60.9%, 694/1140) and lived in rural areas (Table 1).

Concerning serological analysis, 790 individuals had an immunity profile and 336 had a susceptibility profile. IgG antibodies were detected in 804 individuals, resulting in a prevalence of 70.6% (804/1140; 95% CI: 67.9–73.2%) 

Considering only individuals from rural areas, the prevalence was 62.6% (95% CI: 58.9–66.3%). On the other hand, in the urban area, a prevalence of 81.9% was found (95% CI: 78.4–85.4%). According to the bivariate analysis, living in an urban area was a risk factor associated with seropositivity for IgG (*p* < 0.0001), where the risk of being infected was 2.7 times higher (95% CI = 2.0–3.6%).

In the rural area, the bivariate analysis showed a statistically significant difference between seropositivity for IgG and contact with cats (*p* = 0.017) and age group (*p* < 0.0001) (Table 2).

In the multivariate analysis, the seroprevalence of *T. gondii* infection was evaluated in relation to age group, cat contact, açaí intake, hunting meat intake and the habit of entering the forest. The final logistic regression model showed an association between seropositivity for IgG and age group (*p* < 0.0001, OR = 6.2) and cat contact (*p* = 0.003, OR = 1.8) (Table 3).

In the urban area, the bivariate analysis showed a significant difference only for age group (over 20 years old) (*p* < 0.05) (Table 4). In the multivariate analysis, the seroprevalence of *T. gondii* infection was evaluated in relation to age group and water treatment by boiling or filtering. The final logistic regression model showed an association between seropositivity for IgG and age group (*p* = 0.002, OR = 2.1) (Table 3).

An increase in seroprevalence was observed with age (*p* < 0.05). In the rural area, the variation ranged from 28.6% (42/147; 95% CI: 21.3–35.9%) in children aged 1 to 10 years to 92.3% (48/52; 95% CI: 89.3–95.3%) among those over 60 years old. In the urban area, the variation ranged from 69.3% (61/88; 95% CI: 59.7–79.0%) to 87.5% (35/40; 95% CI: 37.6–50.7%), respectively (Figure 2).

## 4. Discussion

The study provided information on the prevalence of *T. gondii* infection in rural and urban populations in the municipality of Ponta de Pedras, where an outbreak of toxoplasmosis occurred and which presents different landscapes and types of exposure. Thus, it was an opportunity to assess population groups and risk factors associated with infection, contributing to knowledge of the *T. gondii* cycle in the Amazon.

In Brazil and, more specifically, in the Amazon, the high prevalence of toxoplasmosis is a reality. Previous studies investigated specific communities or areas, such as indigenous [14], riverside [15] and urban areas [6], and the present study evaluated the prevalence of toxoplasmosis and associated risk factors in individuals living in urban and rural areas, including riverside communities.

The total prevalence in the studied populations was 70.4%, a rate higher than that described in countries such as Scotland, China and the United States [2,3,16] and in Brazilian cities such as Niterói (Rio de Janeiro) and Pelotas (Rio Grande do Sul) [17,18] and below the rates observed in other cities in the state from Pará as Belém and Novo Repartimento [6,19].

However, when analyzed by place of residence, it was observed that the prevalence in the rural area (63.1%) was lower than that in the urban area (81.3%), where the risk of being infected was 2.7 times higher. This difference of greater prevalence in rural areas has often been demonstrated in Brazil [20,21] and in other countries [22,23]. On the other hand, in studies carried out in tropical countries such as Sri Lanka and Burkina Faso, a higher seroprevalence was observed among residents of urban areas, which can be explained by the conditions of poverty, overpopulation and inadequate water supply systems [24,25].

In the Amazon region, a study carried out in an indigenous community in the Alto Rio Negro basin (Amazonas) demonstrated a high frequency of detection of antibodies against *T. gondii*, which was possibly related to the process of acculturation, demographic concentration and urbanization without sanitary/health infrastructure [14]. In Ponta de Pedras, an urbanization process similar to that described above was observed and, in general, it was fast and disordered with environmental degradation, high human density, inadequate infrastructure and precarious access to treated water [26].

The prevalence observed among individuals in rural areas in this study was close to that reported for riverside communities in Lábrea, Amazonas. The authors highlighted some important points, such as persistent rains causing river floods, which make it difficult to raise animals in the home and dilute environmental contamination, reducing the likelihood of contamination of this area by oocysts [15].

In Brazil, the importance of transmission by oocysts in urban areas has already been demonstrated due to the high contamination by this infectious form, the diversity of *T. gondii* genotypes and the absence of actions to control stray cats [27]. In the municipality studied, there is no available information on the presence of cats, although 27.3% of residents of the urban area and 30.2% of rural areas reported having a cat, which demonstrates the presence of the animal in both areas. It is worth mentioning that it is not common in Brazilian Amazon municipalities to keep cats only inside the home and to use litter boxes, and these behaviors can contribute to environmental contamination.

The association between IgG seropositivity and cat contact was only in rural areas, where there is a low density of animals and houses that are far apart and isolated by barriers such as rivers and forests. In the urban area, the homes are close together, and there are high-density cat population groups, which have frequent access to the street, direct contact among the animals and circulation in areas such as fairs, butcher shops and markets.

Infected animals can travel greater distances, release oocysts and expose families that do not have animals to a similar risk of infection [13]. Therefore, having a cat at home was not found to be a risk factor in the rural and urban areas of the Brazilian Amazon, as demonstrated by previous studies [13,28]. The characteristics cited above and the conditions of temperature and humidity that favor the survival and dispersion of oocysts establish high environmental contamination and enable several routes of infection for humans and animals.

The increase in infection prevalence related to age has been demonstrated previously [29,30] and is a result of duration of exposure to the parasite. Seroprevalence studies in individuals up to 20 years of age in countries such as China, Portugal and Angola showed rates between 8 and 18% [31,32]. In the Amazon, in children up to 14 years old, a seroprevalence of 35.5% was observed in residents of the rural area of Acre and 46% in riverside communities in Amazonas [13,15]. In Ponta de Pedras, similar rates were found in children from rural areas, but in urban areas, the prevalence was much higher.

In the analysis of the age prevalence curve, the different epidemiological profiles between urban and rural areas was prominent, mainly in the age group up to 10 years, which presented 28.6% seropositivity in rural areas and 69.3% seropositivity in the urban area. Thus, in the urban area, a high prevalence among children was observed, and no significant increase after that age was observed, in contrast to rural areas, which showed an upward curve. These facts can be explained by the different scenarios since urban areas have a high population density and large circulation of stray cats and rural areas have a low population density and small circulation of stray cats. Therefore, individuals in the urban area probably live in an environment with great contamination by oocysts and contact with the parasite from the first years of life.

Thus, it is possible that the epidemiological profiles observed in this study represent the epidemiology of *T. gondii* infection in municipalities in the Amazon region. The “tropical model” of toxoplasmosis transmission (individuals exposed to an environment highly contaminated by oocysts, with the first parasite contact occurring early in childhood) can be applied only to urban areas of the Amazon and the “Latin model” (seropositivity increases gradually with age, with the first contact with the parasite occurring mainly among young adults) to rural and wild areas [33]. Thus, urban area growth in the Amazon may increase exposure to *T. gondii*, including to atypical strains, in a reverse trend to that observed in other countries, such as France and the United States, where the prevalence has decreased in recent years [3,34].

In different locations, especially in riverside areas, access to health services is difficult, and infection by atypical strains that tend to present unusual and severe symptoms can result in a longer delay in diagnosis and treatment implementation.

## 5. Conclusions

This study showed the epidemiological profiles of *T. gondii* infection in urban and rural areas of a municipality in the Amazon, including remote areas and traditional communities. It was observed that residents of urban areas have greater exposure to the parasite starting in the first years of life, and a transmission pattern that is possibly like that observed in other tropical developing countries. Thus, it is important to establish strategies aiming at planned urban growth with a treated water supply, sanitary structure and control of stray animals. In addition, studies involving the analysis of environmental samples (soil, water, food) and the use of molecular techniques should aid in acute toxoplasmosis surveillance and understanding how environmental changes can affect the transmission of *T. gondii* in the Amazon region.

## Figures and Tables

**Figure 1 ijerph-18-08664-f001:**
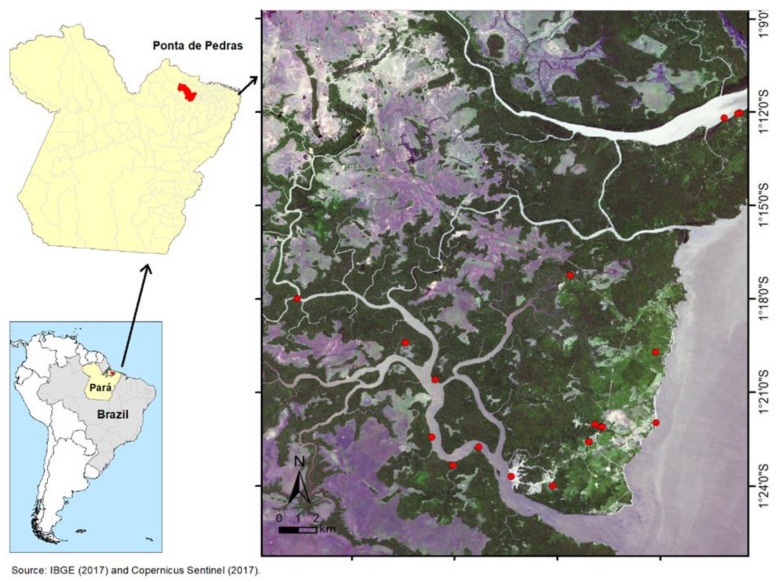
Map of the municipality of Ponta de Pedras, Pará State, Brazil (points in red: collection locations).

**Figure 2 ijerph-18-08664-f002:**
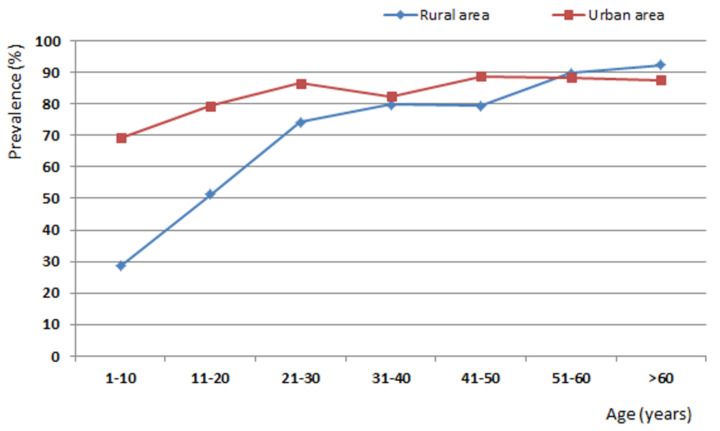
Anti-*T. gondii* IgG prevalence curve by age group in individuals living in urban and rural areas in the municipality of Ponta de Pedras, Pará, Brazil, from August 2015 to April 2018.

**Table 1 ijerph-18-08664-t001:** Variables studied and seropositivity of IgG anti-*T. gondii* in individuals residing in the municipality of Ponta de Pedras, Pará, Brazil, from August 2015 to April 2018.

	Rural Area		Urban Area		Total	
Variables	*n*	IgG Positive (%)	*n*	IgG Positive (%)	*n*	IgG Positive (%)
Sex						
Male	283	185 (65.4)	163	133 (81.6)	446	318 (71.3)
Female	388	235 (60.6)	306	251 (82.0)	694	486 (70.0)
Age group						
1–10	147	42 (28.6)	88	61 (69.3)	235	103 (43.8)
11–20	158	81 (51.3)	87	69 (79.3)	245	150 (61.2)
21–30	113	84 (74.3)	74	64 (86.5)	187	148 (79.1)
31–40	84	67 (79.8)	74	61 (82.4)	158	128 (81.0)
41–50	68	54 (79.4)	63	56 (88.8)	131	110 (84.0)
51–60	49	44 (89.8)	43	38 (88.4)	92	82 (89.1)
>60	52	48 (92.3)	40	35 (87.5)	92	83 (90.2)
Soil contact						
Yes	583	218 (37.4)	284	234 (82.4)	867	452 (52.1)
No	88	55 (62.5)	185	150 (81.1)	273	205 (75.1)
Having a cat at home						
Yes	203	126 (62.1)	128	103 (80.5)	331	229 (69.2)
No	468	294 (62.8)	341	281 (82.4)	809	575 (71.1)
Contact with cats						
Yes	486	318 (65.4)	393	326 (82.9)	879	644 (73.3)
No	185	102 (55.1)	76	58 (76.3)	261	160 (61.3)
River water intake						
Yes	216	132 (61.1)	45	38 (84.4)	261	170 (65.1)
No	455	288 (63.3)	424	346 (81.6)	879	634 (72.1)
Water treatment by boiling or filtering						
Yes	192	118 (61.5)	216	185 (85.6)	408	303 (74.3)
No	479	302 (63.0)	253	199 (78.6)	732	501 (68.4)
Having yard with land or sand						
Yes	635	400 (63.0)	389	323 (83.0)	1024	723 (70.6)
No	36	20 (55.5)	80	61 (76.2)	116	81 (69.8)
Açaí juice intake						
Yes	662	417 (63.0)	454	372 (81.9)	1116	789 (70.7)
No	9	3 (33.3)	15	12 (80.0)	24	15 (62.5)
Raw/undercooked meat intake						
Yes	177	115 (64.9)	130	103 (79.2)	307	218 (71.0)
No	494	305 (61.7)	339	281 (82.9)	833	586 (70.3)
Hunted meat intake						
Yes	448	290 (64.7)	189	152 (80.4)	637	442 (69.4)
No	202	117 (57.9)	280	232 (82.8)	482	349 (72.4)
Habit of entering the forest						
Yes	403	262 (65.0)	176	147 (83.5)	579	409 (70.6)
No	250	149 (59.6)	291	235 (80.7)	541	384 (71.0)

*n*: number.

**Table 2 ijerph-18-08664-t002:** Bivariate analysis between the variables studied and the seropositivity of IgG anti-*T. gondii* in individuals residing in the rural area of the municipality of Ponta de Pedras, Pará, Brazil, from August 2015 to April 2018.

Variables	*n*	IgG Positive (%)	OR (CI 95%)	*p*-Value
Sex				
Male	283	185 (65.4)	1.2 (0.9–1.7)	0.234
Female	388	235 (60.6)		
Age group				
1–10	147	42 (28.6)	Refer.	-
11–20	158	81 (51.3)	2.6 (1.6–4.2)	<0.0001
21–30	113	84 (74.3)	7.2 (4.1–12.6)	<0.0001
31–40	84	67 (79.8)	9.8 (5.2–18.7)	<0.0001
41–50	68	54 (79.4)	9.6 (5.8–19.2)	<0.0001
51–60	49	44 (89.8)	22.0 (8.2–59.3)	<0.0001
>60	52	48 (92.3)	30.0 (10.2–88.4)	<0.0001
Soil contact				
Yes	583	218 (37.4)	1.0 (0.6–1.6)	0.921
No	88	55 (62.5)		
Having a cat at home				
Yes	203	126 (62.1)	0.9 (0.7–1.4)	0.922
No	468	294 (62.8)		
Contact with cats				
Yes	486	318 (65.4)	1.5 (1.1–2.2)	0.017
No	185	102 (55.1)		
River water intake				
Yes	216	132 (61.1)	0.9 (0.6–1.3)	0.645
No	455	288 (63.3)		
Water treatment by boiling or filtering				
Yes	192	118 (61.5)	0.9 (0.7–1.3)	0.767
No	479	302 (63.0)		
Having yard with land or sand				
Yes	635	400 (63.0)	1.4 (0.7–2.7)	0.471
No	36	20 (55.5)		
Açaí juice intake				
Yes	662	417 (63.0)	3.4 (0.8–13.7)	0.139
No	9	3 (33.3)		
Raw/undercooked meat intake				
Yes	177	115 (64.9)	1.1 (0.8–1.6)	0.502
No	494	305 (61.7)		
Hunted meat intake				
Yes	448	290 (64.7)	1.3 (0.9–1.9)	0.116
No	202	117 (57.9)		
Habit of entering the forest				
Yes	403	262 (65.0)	1.3 (0.9–1.7)	0.191
No	250	149 (59.6)		

*n*: number; OR: odds ratio; CI: confidence interval; Refer: reference.

**Table 3 ijerph-18-08664-t003:** Final model of multiple logistic regression between the studied variables and anti-*T. gondii* IgG seropositivity in individuals residing in rural and urban areas of the municipality of Ponta de Pedras, Pará, Brazil, from August 2015 to April 2018.

Variable	OR	CI 95%	*p*-Value
Rural area			
Age (≥21years )	6.217	4.33–8.92	<0.001
Contact with cat	1.799	1.22–2.65	0.003
Urban area			
Age (≥21 years )	2.118	1.31–3.41	0.002

OR: odds ratio; CI: confidence interval.

**Table 4 ijerph-18-08664-t004:** Bivariate analysis between the studied variables and anti-*T. gondii* IgG seropositivity in individuals living in the urban area of the municipality of Ponta de Pedras, Pará, Brazil, from August 2015 to April 2018.

Variables	*n*	IgG Positive (%)	OR (CI 95%)	*p*-Value
Sex				
Male	163	133 (81.6)	0.9 (0.6–1.6)	0.991
Female	306	251 (82.0)		
Age group				
1–10	88	61 (69.3)	Refer.	-
11–20	87	69 (79.3)	1.7 (0.8–3.7)	0.180
21–30	74	64 (86.5)	2.8 (1.3–6.3)	0.016
31–40	74	61 (82.4)	2.1 (0.9–4.5)	0.073
41–50	63	56 (88.8)	3.5 (1.4–8.7)	0.0083
51–60	43	38 (88.4)	3.4 (1.2–9.5)	0.030
>60	40	35 (87.5)	3.1 (1.1–8.8)	0.047
Soil contact				
Yes	284	234 (82.4)	1.09 (0.7–1.8)	0.811
No	185	150 (81.1)		
Having a cat at home				
Yes	128	103 (80.5)	0.8 (0.5–1.5)	0.726
No	341	281 (82.4)		
Contact with cats				
Yes	393	326 (82.9)	1.51 (0.8–2.7)	0.225
No	76	58 (76.3)		
River water intake				
Yes	45	38 (84.4)	1.22 (0.5–2.8)	0.789
No	424	346 (81.6)		
Water treatment by boiling or filtering				
Yes	216	185 (85.6)	1.61 (0.9–2.6)	0.066
No	253	199 (78.6)		
Having yard with land or sand				
Yes	389	323 (83.0)	1.5 (0.8–2.7)	0.202
No	80	61 (76.2)		
Açaí juice intake				
Yes	454	372 (81.9)	1.1 (0.3–4.1)	0.801
No	15	12 (80.0)		
Raw/undercooked meat intake				
Yes	130	103 (79.2)	0.8 (0.5–1.3)	0.431
No	339	281 (82.9)		
Hunted meat intake				
Yes	189	152 (80.4)	0.8 (0.5–1.4)	0.583
No	280	232 (82.8)		
Habit of entering the forest				
Yes	176	147 (83.5)	1.2 (0.7–1.9)	0.531
No	291	235 (80.7)		

*n*: number; OR: odds ratio; CI: confidence interval; Refer: reference.

## Data Availability

The data presented in this study are available upon reasonable request from the corresponding author.

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
