# Peer review of "T. gondii Infection in Urban and Rural Areas in the Amazon: Where Is the Risk for Toxoplasmosis?"

_ijerph, 2021, doi:10.3390/ijerph18168664_

Round 1

Reviewer 1 Report

The manuscript is well written and easy to read and follow the text!

However, there are minor things that should be considered:

L13 and L160 - Authors mention that there had been registered an outbreak of toxoplasmosis. It would be very interesting to know more details regarding this outbreak - was it from acquired or congenital toxoplasmosis? How was it detected? What was the source of infection?

Materials and Methods

L90 – What was the reason to exclude pregnant women from the study?

L99 – In this section authors mentions that they also test for the presence of IgM antibodies, however they do not mention obtained results in the Result section. That is why I do not see the reason to mention this in the M&M section.

Results

L112 and L115 – Duplicates the information regarding rural area.

Discussion

L173-175 – These results (specifically odds ratio) have not been previously mentioned in the result section. I would suggest to move this to Result section.

Author Response

Thank you

Kind regards

Reviewer 2 Report

The manuscript entitled T. gondii infection in urban and rural areas in the Amazon: 2 where is the risk for toxoplasmosis? “ is an important contribution on the knowledge of Toxoplasma circulation in a specific exposed population, after an outbreak occurrence. The overall quality of the manuscript is of high value, only minor concerns are listed below.

Data about interviews should be better detailed (maybe as supplementary material?or a summarizing table?)

Please indicate the sensitivity and specificity of the ELISA kit used ELISA) (Symbiosis Diagnóstica Ltda., 100 Leme, Brazil).

Line 106-107 “variables that had a p-value < 0.200 were performed by the stepwise forward method” could you please explain better this threshold used as pvalue 0.2?

T. gondii in italics along all ms

Author Response

Thank you

Kind regards
